# Selective Passive/Active Switchable Knee Prosthesis Based on Multifunctional Rotary Hydraulic Cylinder for Transfemoral Amputees

**Hyunjun Shin, Jinkuk Park, Huitae Lee, Sungyoon Jung, Mankee Jeon and Sehoon Park \***

Advanced Prosthesis R&D Team, Korea Orthopedics & Rehabilitation Engineering Center, 10 Beon-gil, Gyeongin-ro, Bupyeong-gu, Incheon 21417, Republic of Korea; hjshin@comwel.or.kr (H.S.); korecdoggabi7@comwel.or.kr (J.P.); leehuitae@kcomwel.or.kr (H.L.); syjung0410@comwel.or.kr (S.J.); mkjeon@comwel.or.kr (M.J.)
\* Correspondence: mbb1020@kcomwel.or.kr

**Abstract:** Significant advances have been made in prostheses with the aim of enhancing the quality of life for transfemoral amputees. While commercially available transfemoral prostheses mainly focus on the developing passive prostheses that act only as dampers, academic research is centered round powered prostheses that can provide net-positive knee torque. Although recent active-powered prostheses have made excellent progress in terms of weight and battery life, it remains unclear if these prostheses can be effectively used in everyday life. This study presents a rotary hybrid prosthesis based on the combination of a multifunctional rotary hydraulic cylinder that is designed to operate as a brake, clutch, and damper with a 100 W active motor system. This prosthesis enables long-term level ground walking while supplying active power as needed. The passive and active components of this hybrid prosthesis are designed such that they can be decoupled when operated independently, allowing for quick switching between passive and active modes in 50–100 ms. The study outlines the aims and procedures for the design of rotary hybrid prostheses, as well as the feasibility tests for each module and the amputee's clinical test on the developed knee prosthesis.

**Keywords:** knee prosthesis; multifunctional rotary cylinder; selective mode; hydraulic nozzle; transfemoral amputee; powered prosthesis

## 1. Introduction

A transfemoral prosthesis is a device that helps amputees walk by replacing bodily functions. Over 185,000 people with lower limb amputees use prostheses on a daily worldwide [1]. The function, weight, and convenience of the lower limb prosthesis have a significant impact on the quality of life of lower limb amputees. The knee prostheses may be categorized as purely passive, electrically controlled passive, or fully active according to the driving method [2–5]. Majority of the commercial items are either purely passive or electrically controlled passive type. The typical weight of such items is within the range of 1.65 kg, which allows them to be continuously used for more than one day, and the person can walk on level ground or mild hills without any discomfort. However, because current commercial passive prostheses cannot generate net-positive energy, their ability to walk on ascending stairs or steep slopes is limited. Therefore, recent studies on prostheses are aimed at the generation of active power to lift the human body.

In 2006, Professor Goldfarb's research team at Vanderbilt University developed a low-extremity prosthesis with a maximum torque of 86 Nm using a pneumatic pump, and in 2008, they developed a low-extremity prosthesis with a maximum torque of 75 Nm utilizing a 150 W BLDC motor and a ball screw. After 2012, the design was modified to include a 200 W class BLDC motor and a three-stage pulley-belt type, and a prosthesis with a maximum torque of 85 Nm was developed accordingly [6–11]. These studies demonstrated

that an active prosthesis could aid amputees in walking on stairs or steep slopes. Professor Herr's research team at MIT University used a series of elastic mechanisms and cable pulleys to build a knee prosthesis that can provide 130 Nm of torque and 58 RPM in 2011. The MIT knee inherited the existing series elastic mechanism and used a 200 W class BLDC motor in 2014. The model demonstrated a peak output performance of 120 Nm and a maximum speed of 112 RPM, thus opening up the potential for the commercialization of an active prosthesis [12–14]. The Open Source Leg, developed at the University of Michigan, weighs 2.3 kg and can deliver a torque up to 120 Nm even with a low reduction ratio (42:1) using an aeronautical rotor. Another benefit of this model is that the actuator can be assembled in a short time. This prosthesis is a fully active prosthetic limb that is capable of producing a knee joint torque of over 80 Nm at a speed of more than 40 RPM (240 deg/s), indicating that an amputee may be able to walk on stairs and steep slopes [15]. However, due to their heavy weight and limited use time, these prostheses have not achieved full commercialization. Most amputees prefer a low weight that causes less fatigue, even if the function of the prosthetic limb is limited, and they prefer at least one day of use time on a single charge. Hence, people are often hesitant to adopt a fully active prosthetic limb that weighs more than 4 kg and provides only 1–2 h of battery life.

In order to overcome these drawbacks, recent research is primarily focused on reducing weight and increasing use time. To achieve the above goals, passive components such as springs and hydraulic cylinders have been actively exploited in recent studies. Walking on level ground or low slopes is the norm for ordinary amputees, with steep slopes or stairs being rare. Therefore, to conserve energy, a passive/active module that is capable of selectively driving based on walking conditions must be developed. A slide-crank structure transmission mechanism and a small linear hydraulic cylinder are used in the lightweight knee prosthesis recently developed by the University of Utah and Shirley Ryan AbilityLab. The prosthesis weighs only 1.7 kg, making it almost as light as a typical passive prosthesis. Despite its low weight, it boasts a high torque of 125 Nm and a high speed of 90 RPM, making it a promising candidate for rapid commercialization [16,17]. Their recent study has shown that amputees using advanced robotic leg prostheses could walk on the level ground indefinitely without needing to recharge the battery [18]. Researchers have also developed new drive mechanisms that could cater to both the stance phase, which requires large torque and the swing phase, which requires fast angular velocity. This has been achieved through the use of a two-speed transmission [19] or by connecting two motors in parallel [20], resulting in lighter and more compact prosthesis. Another practical approach is the development of a swing assisted prosthesis that combines a ball screw and a hydraulic cylinder, offering clinical advantages such as longer usage time and reduced weight. [21] Wang et al. developed a hybrid prosthetic leg by connecting tiny hydraulic cylinders and ball screws. The linear hydraulic cylinder, which is a passive module, and the drive section of the active driving unit are decoupled in both mechanical structures [22]. It is feasible to drive the active module alone in the passive mode without the need for an external load to drive the reverse, and this helps save considerable energy.

This study developed a hybrid prosthesis by parallelly linking a rotary hydraulic cylinder, which is a passive element, and a motor-gear module, which is an active element using a timing belt. The driving parts of the active module and passive module may be decoupled by controlling the nozzle in the rotary hydraulic cylinder. The overall energy consumption can be reduced by walking in a passive mode based on the utilization of only the rotary hydraulic cylinder on level ground surfaces and low slopes that do not require active power. Furthermore, when active power is needed, such as while walking on steep slopes or stairs, walking with the help of an active motor module is possible. The rotary hybrid actuator's unique design method and control algorithm are presented in Section 2 of this study. The passive module and actuator drive are evaluated, and the amputee's clinical assessment is introduced in Section 3.

## 2. Methods

### 2.1. Driving Principle of Rotary Hybrid Actuator

The rotary hybrid prosthesis rotates the knee joint using two approaches: one with the rotary cylinder body fixed and the blade within the rotary hydraulic cylinder rotating, and the other with the rotary blade fixed and the rotary hydraulic cylinder body itself rotating. The rotation method is decided according to the amputee's gait situation, and each gait mode can be switched in real-time by adjusting the hydraulic nozzle in the rotary cylinder.

The rotation of the motor drive module is locked in passive mode by the electrical brake of the motor drive module, which prevents the rotation of the rotary cylinder body connected to the timing belt, as illustrated in Figure 1A. As a result, the rotation of the blade within the rotary cylinder is entirely responsible for knee flexion/extension. The rotational resistance of the knee joint can be adjusted in real-time by changing the flow rate of the two-channel flexion/extension flow path that is formed in the rotary hydraulic cylinder through the servo motor and the rotary sleeve nozzle. When operated passively, the rotary hybrid prosthesis functions similarly to a conventional microprocessor-controlled knee (MPK) and requires just two small DC motors to be controlled, thereby minimizing overall energy consumption.

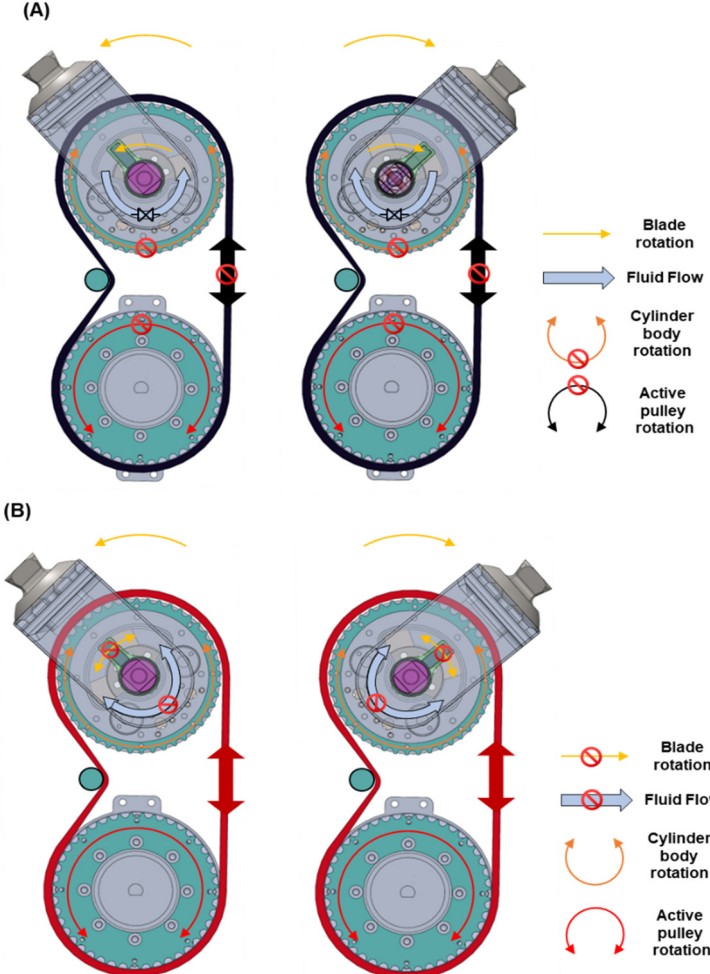

**Figure 1.** Driving rotary hydraulic cylinder and active motor module according to passive/active mode: (**A**) bi-directional rotation of the rotary blade in passive mode, (**B**) bi-directional rotation of cylinder body in active mode.

As shown in Figure 1B, in active mode, the hydraulic nozzle fully closes the flexion/extension nozzle flow path, suppressing the rotation of the rotary blade in both flex-

ion/extension directions. The electronic brake in the motor drive module is released, allowing the motor rotate freely. In addition, the motor's power is responsible for the rotation of the rotary hydraulic cylinder's body through the harmonic gear, pulley, and timing belt, subsequently enabling the rotation of the knee joint.

*2.2. Development of Multifunctional Rotary Hydraulic Cylinder*

2.2.1. Design of Rotary Hydraulic Cylinder

A multifunctional rotary hydraulic cylinder with various functions such as brake, clutch, and damper was developed in this study. The rotary hydraulic cylinder comprises fluid chambers, a blade, and a servo valve, as illustrated in Figure 2A. The fluid chamber is separated into flexion and extension chambers by a blade placed into chamber. The flow paths in the flexion and extension directions are separate, and the fluid flow in the opposite direction is stopped by a ball check valve.

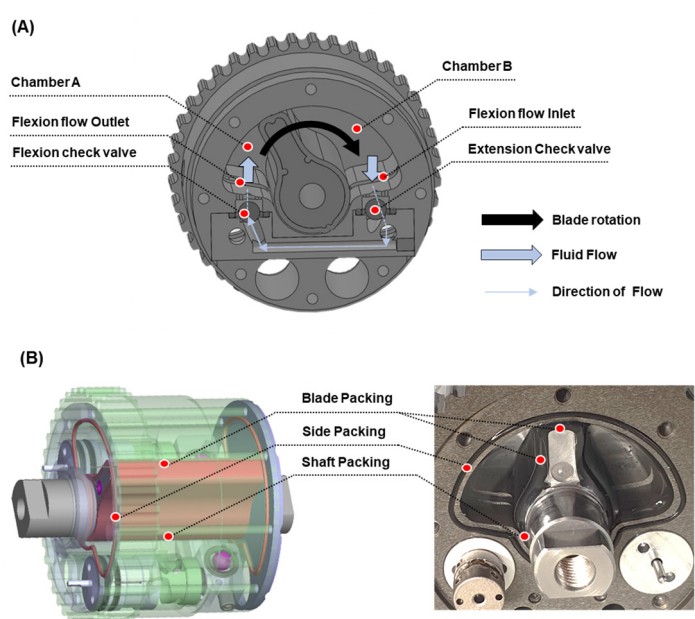

**Figure 2.** Design of rotary hydraulic cylinder: (**A**) flow path structure of rotary hydraulic cylinder and fluid flow during knee flexion; (**B**) 3D modeling of packing in the rotary hydraulic cylinder and actual packing structure.

Figure 2A shows the flow route in the flexion direction. A servo valve equipped with a rotary sleeve nozzle, controlled by a DC motor, is located at the center of the flexion/extension flow channel, and the flow rate of each flow path can be regulated by adjusting the angle of the nozzle. The fluid flow between the two chambers may be stifled if the nozzle is entirely closed, causing the blade to function a brake and clutch, as well as a damper when the nozzle is partly open. Sealing is important for the accurate control of the rotary hydraulic cylinder's different functions. Synchronous rotation of the blade and cylinder body when fluid is flowing between the flexion/extension chambers in the rotary hydraulic cylinder makes it challenging to control the knee angle. Precision control is only achievable when the blade's rotation is totally inhibited during active modes. The primary aim of sealing is to reduce fluid flow by placing rubber packing between the blade and case and between the blade's axis and the case. If the rubber packing is applied too forcefully at this point, the friction of the blade rises significantly, and this results in difficulty in rotation during the swing phase. Therefore, it should be designed such that it inhibits the fluid flow between chambers while reducing rotational resistance. Figure 2B shows the overall 3D modeling structure of the packing and the actual packing structure. The packing between the tip of the circumference of the blade, shaft, and the case produces an assembly height

difference of 0.1 to 0.2 mm, which is tightly sealed by the side cap, generating a fluid flow gap that helps decrease frictional resistance.

### 2.2.2. Shape Design of the Rotary Blade

The rotary cylinder acts as a damper in passive mode and supports the rotation of the knee joint due to the body weight. Fluid pressure is exerted along the blade surface during flexion and extension of the knee joint, as illustrated in Figure 3A. The rotary blade must be optimally designed such that it can bear the design pressure. Figure 3B shows a simplified diagram of a scenario wherein the pressure is applied to the blade. As illustrated in Figure 3C, the red color indicates the area under the blade's pressure, while the black diagonal line represents the shaft area's pressure.

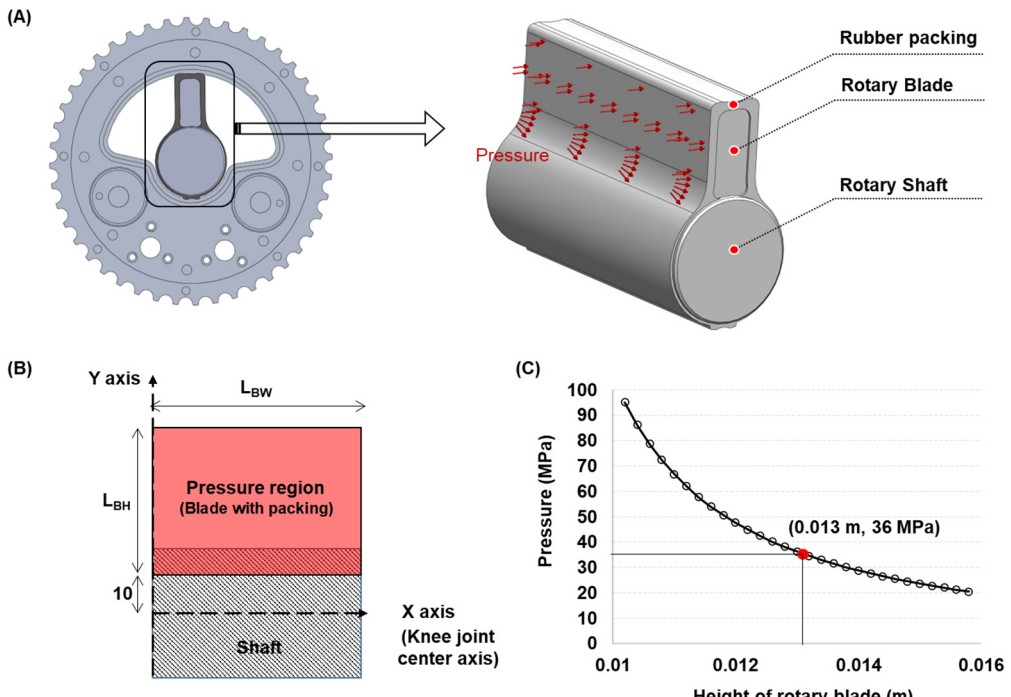

**Figure 3.** Design of rotary blade: (**A**) pressure distribution shape of a rotary blade; (**B**) area where pressure is applied to a rotary blade; (**C**) pressure graph generated according to changes in the height of rotary blade (red dot: optimal shape design poi).

Because the knee moment created by pressure applied to the blade grows in size as it moves away from the center, the moment generated on the whole blade can be expressed using Equation (1).

$$M_{knee} = P_{RB} \cdot \int_0^{L_{BW}} \int_{0.008}^{L_{BH}} y \cdot dy \cdot dx \qquad (1)$$

$$M_{knee} = 0.5 \cdot P_{RB} \cdot L_{BW} \cdot \left( L_{BH}^2 - 0.008^2 \right) \qquad (2)$$

In Equation (1), $M_{knee}$ on the left side denotes the torque applied to the knee center by the body weight, and the right side denotes the damping torque generated by the rotary blade. These values must be equal to receive support from the knee prosthesis without bending the knee. $P_{RB}$ on the right side denotes the pressure applied to the blade surface, $L_{BW}$ denotes the width of the blade, and $L_{BH}$ denotes the height of the blade. In the case of the blade height range of 0–8 mm, a rotational moment is hardly generated because the pressure is directed toward the center of the knee. Therefore, the knee torque was calculated by simplifying it as in Equation (2), considering only the section above 8 mm. Given the maximum extension torque of the bottom active module, the maximum brake torque that the rotary blade should be set at 80 Nm. It was observed that the rubber packing

in the hydraulic cylinder, which was previously developed by our team, was distorted during prolonged operation at a pressure of approximately 40.0 MPa. Thus, the geometry of the blade was chosen in a way that the maximum pressure was 36 MPa when a torque of 80 Nm is applied. Because the height of the blade has a relatively high effect on the knee moment than the width of the rotary blade, the width was set at 42 mm, and the height was altered, as indicated in Equation (2). As shown in Figure 3C, from the calculation of the pressure obtained by altering the height of the rotary blade, the height of the blade against a torque of 36 Nm was determined to be 13 mm. The thickness of the blade, which does not induce deformation, was established by structural analysis, and its value was 5 mm (data not shown). It was also feasible to calculate the blade shape of the smallest size that assured safety by using this optimal design.

### 2.2.3. Structure of Rotary Hydraulic Nozzle

Two rotary hydraulic nozzles are inserted to regulate flexion and extension, which are controlled by two servomotors (DCX12s, Maxon Corp., Switzerland) with a 26:1 gear ratio and a 128-bit encoder. The rotary hydraulic nozzle and servo motor parts are arranged parallelly into the rotary cylinder body and joined at a gear ratio of 1:1 through a pair of spur gears (z = 48, M = 0.3, Pitch = 14.4), as illustrated in Figure 4A.

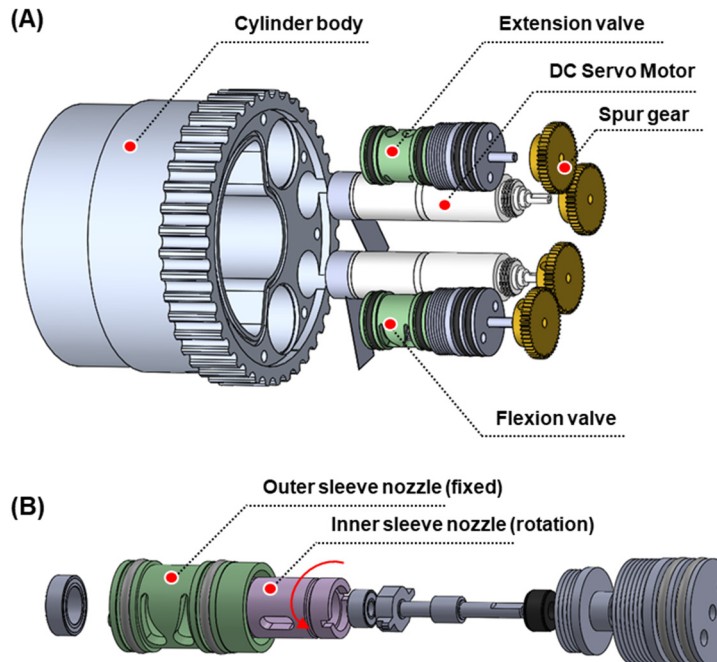

**Figure 4.** The shape of the rotary hydraulic nozzle: (**A**) composition of flexion/extension servo valve; (**B**) composition of the nozzle part. The outer nozzle is fixed, and the inner nozzle is rotated by the motor. Fluid flows through an intersection of the slots of the inner and outer nozzles.

The rotary hydraulic nozzle is a key element that enables the rotary cylinder to function as a brake, clutch, and damper. The construction of the rotary sleeve nozzle changes the fluid flow area during relative rotation between the inner and outer nozzles. As illustrated in Figure 4B, the outer sleeve nozzle is stationary in this study, whereas the inner sleeve nozzle spins; the intersection area between the two nozzles varies. When the cross area is the largest, the state is 'OPEN' with no knee joint resistance; when there is no cross area, the state is 'LOCK', and when it partly crosses, the state is 'DAMPED'. In the swing state, the 'OPEN' state is usually utilized. In typical level ground gait, the 'DAMPED' and 'OPEN' states are suitably tuned to the gait cycle. The 'LOCK' state is employed in the active mode when the motor drives the prosthetic limb, and as stated previously in Section 2.1, the blade is stationary in this situation, and the cylinder body is rotated by the motor drive module.

### 2.3. Motor Drive Module

The motor drive module is connected to the rotary hydraulic cylinder in parallel, and the power of the motor is increased because of the gear ratio between the harmonic gear and the pulley. In passive mode, the motor drive module is locked, and it works like a case frame. In active mode, the motor's driving power is delivered in the upward direction via the pulley and timing belt in order to rotate the hydraulic cylinder body. A spring-actuated-type electrical brake controls the changeover between passive and active modes (BXR-040-10LE, Miki pulley Corp., Japan). In the 'OFF' state, the electrical brake prevents the power of the motor from being sent to the hydraulic cylinder at the top. When active power is required, the electrical brake is released, and the motor's power is sent to the hydraulic cylinder at the top. Given that most amputees walk on level ground or low slopes, the default mode is a condition wherein power is not delivered to the active drive module. The electronic brake is directly attached to the rotor of a frameless motor (6013B, Kollmorgen Corp., USA), which is the rotating part on the center axis to produce the power unit and the clutch unit, as illustrated using the cross-sectional view in Figure 5. The end of the central axis is directly attached to the wave generator of the 50:1 harmonic gear (SHD-14-50-2SH, Harmonic Drive Corp., Japan) and is utilized as the harmonic gear's input. At this point, the outer CRB outer ring is secured to the case frame, and power is delivered to the lower pulley through the harmonic gear's Flexspline as an output. A GT5 standard timing belt was used to link the upper and lower pulleys at a 1:1 ratio (GBN415EV5GT-90, MISUMI Corp., Japan). A belt width of 9 mm was chosen based on the maximum tension and speed of the belt. The gear ratio of the top and lower pulleys varies based on the patient's gait characteristics. An active torque of roughly 40 Nm can be produced in the gear ratio of 1:1, while an active torque of up to 65 Nm can be produced in the gear ratio of 1:1.7.

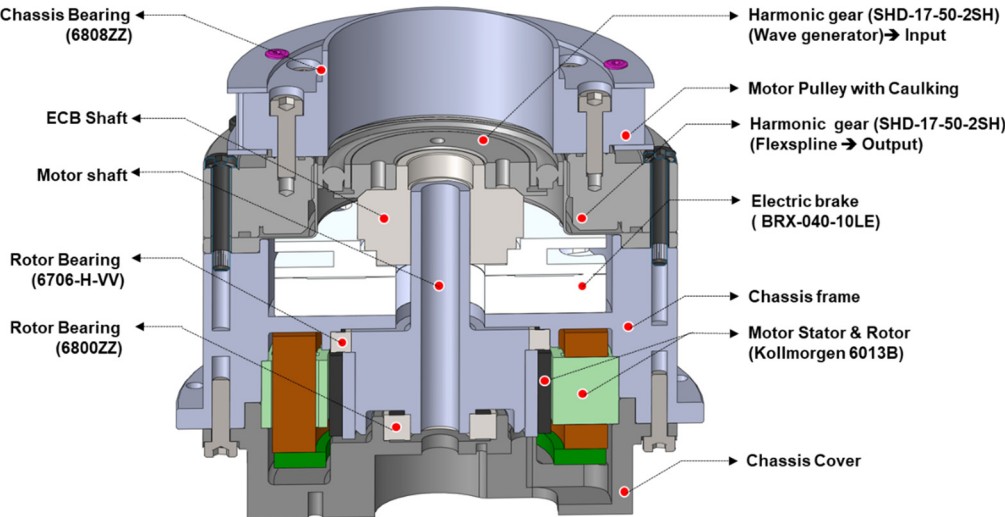

**Figure 5.** Cross-sectional view of the active drive module.

### 2.4. Sensor and Frame Modules

The complete frame, illustrated in Figure 6A, is composed of a sensor module that determines an amputee's gait condition and a three-tiered casing module. Four types of sensors have been incorporated into this hybrid prostheses frame: a 150 Nm class torque sensor (CAS Corp., Korea) that measures knee moment, a strain gauge type ankle moment sensor that measures ankle moment, a potentiometer type knee angle measurement sensor (WAL200, Contelec Corp., Switzerland), and a potentiometer (WAL200) that can measure the relative angle between the rotary hydraulic cylinder body and the blade. The top frame is connected to the rotary cylinder blade through the torque sensor and to the socket via the pyramid head. The ankle moment sensor and pyramid head are used to connect the

lower frame to the ankle prosthesis. The rotary hydraulic cylinder and motor drive module are 92.1 mm away from the main frame, and power is transmitted through pulleys and timing belts. Furthermore, the interior of the main frame case is concave, which allows the knee prosthesis to secure more than 120° of range of motion (ROM). An idler has been attached to the interior portion to allow the timing belt to be driven down the concave surface. Figure 6B shows the manufacturing of the final mechanism, and Table 1 shows the components in each section of the final mechanism.

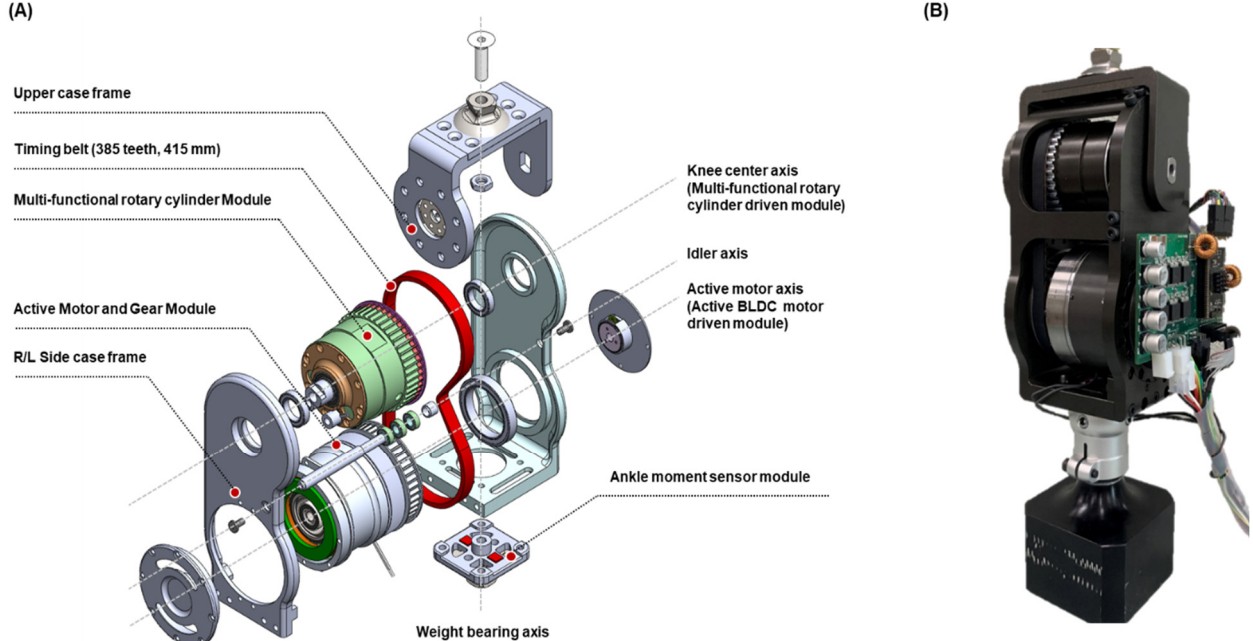

**Figure 6.** Overall system structure: (**A**) parts diagram of the final prototype; (**B**) final prototype.

**Table 1.** Main parts list of rotary hybrid prosthesis.

| Part | Specification |
|---|---|
| Active Motor | Kollmorgen, TMBS-6013B |
| Rotary Nozzle motor | Maxon, DCX12s with 26:1 planetary gear and CPT1024 encoder |
| Harmonic gear | Harmonic Drive, SHD14 |
| Pulley | Manufactured, Gear ratio 1:1, 72 teeth |
| Timing belt | MISUMI, EV5GT, 83 teeth, 415 mm |
| Electrical Brake | MIKI PULLEY, BXR-040-10LE |
| | IMU: EBIMU-9DOF |
| Sensors | Knee angle sensor: WAL200/Rotary case angle sensor: WAL200 |
| | Strain gauge type ankle moment sensor |
| | Knee moment sensor: CAS torque sensor |

## 2.5. Embedded Electronics

Figure 7 shows the four levels of the integrated controller. The power system at the top was developed to provide high torque in the active driving section by delivering an output current of up to 50 A. The primary controller was configured using a TMS28335 chip and a 400 MIPS speed high-end processor from TI, which drove the gait cycle discrimination algorithm calculated at a period of 100 Hz and controlled the active actuator and nozzle

servo valves at 20 kHz. This integrated board uses the values of eight sensors (two DC motor encoders, two potentiometers, two loadcell channels, a torque sensor, and an IMU) to detect the amputee's gait state in real-time. Following this, the motor and electrical brake are given the ideal control value for the present walking condition as input.

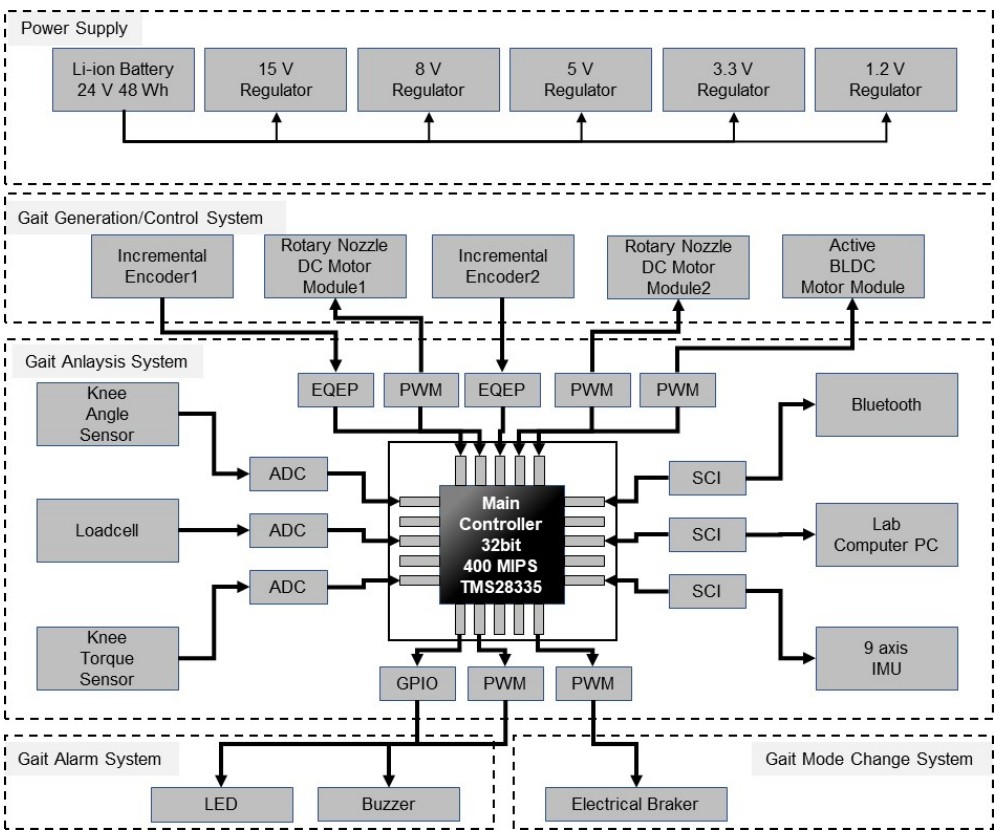

**Figure 7.** Overview of embedded system architecture.

*2.6. Control Algorithm*

2.6.1. Gait Event Detection

Unlike people without disabilities, amputees have limited stance flexion (or knee bouncing) and are hesitant while exerting stress on their prostheses. Thus, the stance phase on the sound side is longer than that on the prosthetic limb side, resulting in an uneven gait pattern. Because the gait cycle and gait speed are irregular compared to people without disabilities, it is important to detect the gait event precisely. Based on the readings of the four sensors, this prosthesis detects the amputees' current gait cycle (early stance, terminal stance, swing flexion, and swing extension) at a sampling rate of 100 Hz (knee angle, IMU, ankle moment, and knee moment sensors). The knee torque sensor is utilized solely for active gait control, such as stair climbing, and it is not used for passive walking control.

Figure 8 shows the detection criteria for each gait period during level-ground walking, which is the same for passive and active modes. The gait event is detected using the hip angle, knee angle, and ankle moment sensor readings from the IMU sensor for the load response and heel rise (toe-off) events, which have a significant impact on gait stability. The knee angle and knee angular velocity are used to identify mid-swing, which is critical for gait speed control. The threshold value for event extraction varies depending on the amputee's gait situation and gait ability, and it is set via a pre-test before wearing the prosthesis.

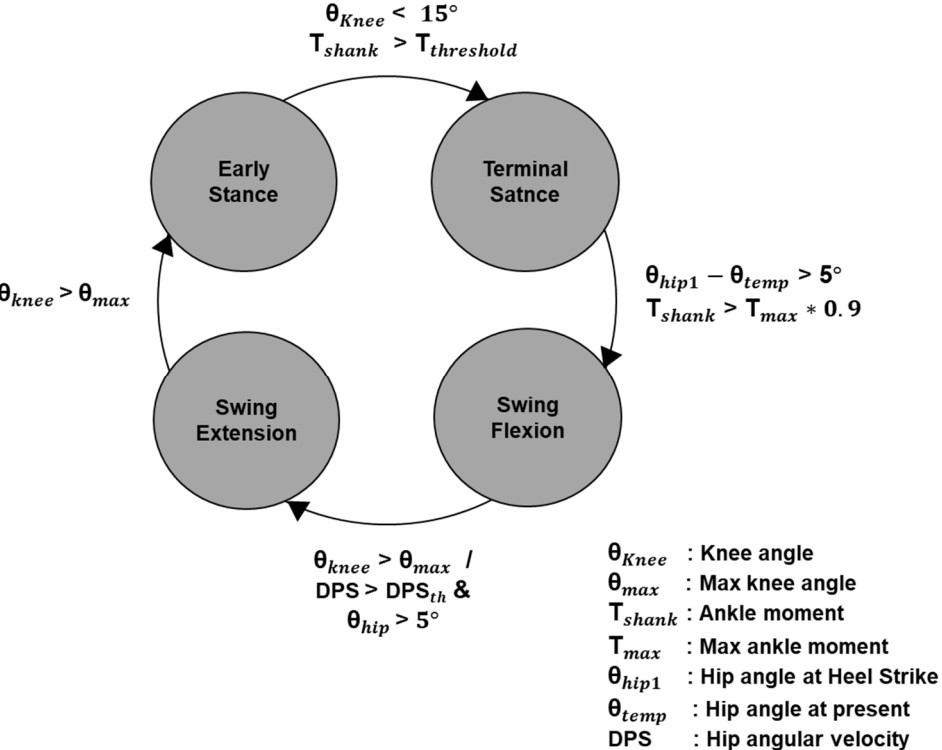

**Figure 8.** Walking state machine for passive and active walking.

### 2.6.2. Gait Control in Passive Mode

When the rotary hydraulic cylinder enters passive mode, the angle between the body and the blade is adjusted such that the blade is shifted to the left end of the hydraulic cylinder. Next, the electrical brake of the active drive module is locked to prevent the cylinder body rotation. In the passive mode, the ideal hydraulic damping value of the hydraulic cylinder for each gait cycle is generated based on the event detection in the previous section by modifying the nozzle angle value in the cylinder. The most appropriate knee joint resistance value for each patient, that is, the nozzle angle value, is established through a pre-test before wearing. The angle of the flexion nozzle is kept in the range of 30–35° throughout the stance phase to ensure that the knee is not bent, while a modest stance flexion is created. It stays open at 10° below after the heel rise point for swing phase ambulation. During the mid-swing phase, a nozzle angle of 50° or higher is maintained to ensure that the knee angle does not exceed 70°, and these conditions help regulate the gait speed. After keeping the extension nozzle at 50° or more to avoid collision in the terminal swing section, it is opened now of heel contact to completely extend the knee.

### 2.6.3. Gait Control in Active Mode

When the rotary hydraulic cylinder enters the active mode, the flexion and extension nozzles are 'closed' to block the fluid flow and fix the rotary blade. The electrical brake of the active driving module unit is then released, causing the rigid body cylinder body to revolve via the motor. In the active mode, gait control appropriate for each gait cycle is conducted using motor torque according to the event detection outlined in the preceding section. A swing phase gait pattern with a maximum bending angle of 60–65° is developed using a position control method in level ground walking. When a torque over the specified threshold is measured on a steep slope or stairwell in the torque sensor, the active motor is employed during the stance phase to provide the extension torque in the direction of raising the weight.

### 2.7. Construction of Test System

In this study, we developed a 2-link type jig that can be connected to the Instron Universal Testing Machine (E10000) to detect knee-damping torque, as shown in Figure 9A. The vertical link is attached to the piston part of the intron, and the joint arm is connected to the prosthetic limb's pyramid head. The revolute joint secures the vertical link and the joint arm. This mechanism may be modeled as a slider-crank system. A schematic diagram of the computation of the damping torque is shown in Figure 9B. Table 2 shows the information for each variable and link length. The knee angle is determined when the Instron piston advances vertically using $d_{inst}$, as shown in Equation (3). At this point, the torque $T_{inst}$ produced at the knee is determined, as shown in Equation (4). In order to establish an equilibrium condition wherein the knee does not bend, the knee torque computed by Equation (4) and $T_d$ produced by nozzle control of the actual prosthesis must be the same. We compared the above results using a digital torque wrench to verify the damping torque measured in the Instron system.

$$\theta = 90 - \cos^{-1}\left( \frac{L_{jarm}^2 + L_{jig}^2 - L_{ver}^2}{2 \cdot L_{jarm} \cdot L_{jig}} \right) \tag{3}$$

$$
\begin{aligned}
T_{inst} &= T_d \\
&= L_{arm} \cdot F_{inst} \cdot \cos\left( \cos^{-1}\left( \frac{L_{ver}^2 + L_{jig}^2 - L_{jarm}^2}{2 \cdot L_{ver} \cdot L_{jig}} \right) \right) \cdot \sin\left( \cos^{-1}\left( \frac{L_{jarm}^2 + L_{jig}^2 - L_{ver}^2}{2 \cdot L_{jarm} \cdot L_{jig}} \right) \right)
\end{aligned} \tag{4}
$$

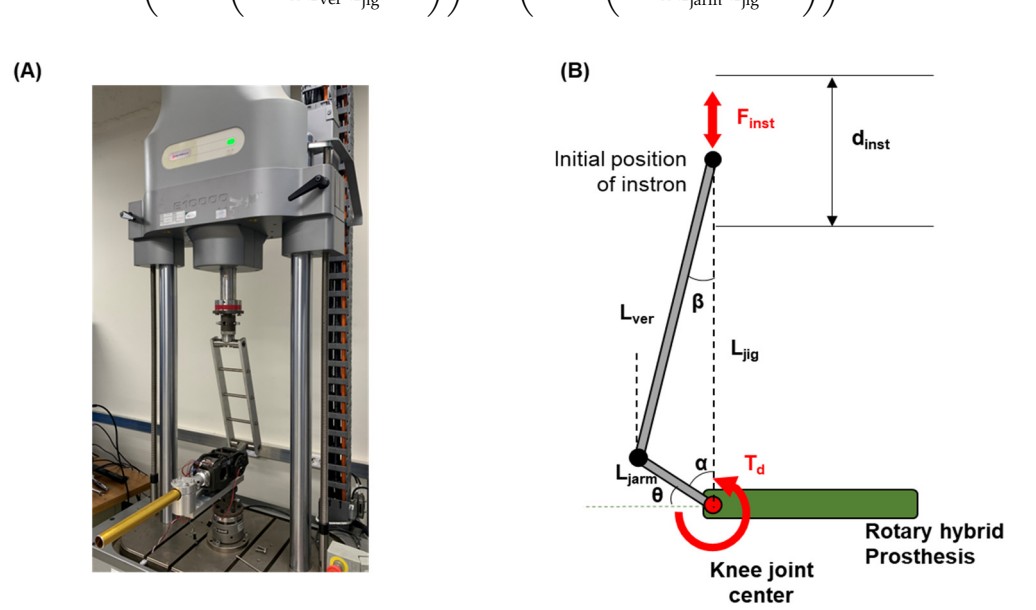

**Figure 9.** (**A**) Measurement test environment of damping torque; (**B**) calculating damping torque through Instron force and displacement measurement.

**Table 2.** Parameters for damping torque calculation model.

| Variable | Value | Description |
|---|---|---|
| $F_{inst}$ | Measured | Loading force by Instron |
| $d_{inst}$ | Measured | Vertical displacement of Instron |
| $T_d$ | Calculated | Damping torque on knee center |
| $\theta$ | Calculated | Relative knee angle |
| **Design Constant** | **Value** | **Description** |
| $L_{ver}$ | 386.8 mm | Length of jig vertical bar |
| $L_{jarm}$ | 400 mm | Length of jig arm |
| $L_{jig}$ | $101 + d_{inst}$ | Length between knee center and initial position of Instron |

## 3. Results

### 3.1. Verification of Passive Mode Operation (Feasibility Test)

The following requirements must be met to achieve a sufficient performance of the rotary hybrid prosthesis in passive mode. First, the active drive module must be locked such that the rotary cylinder body does not move. When the cylinder body and the blade move simultaneously, it is difficult to control the knee joint resistance, which makes it almost impossible to control the knee angle in accordance with the gait cycle. When the electrical brake was turned off, the active drive module's maximum braking torque was measured to be 65.1 Nm (data not shown). This is almost identical to the number that is obtained by multiplying the maximum static frictional torque of the electrical brake, 1.32 (Nm), with the harmonic gear ratio, 50:1. This value represents sufficient braking torque to immobilize the cylinder body while walking normally in passive mode. Second, the transition from the stance phase to the swing phase must occur within 100 ms to minimize the unstable time during which the weight support is withdrawn, while the bent knee allows steady walking.

Figure 10A shows that while converting from the stance phase to the swing phase, the nozzle angle, which was initially 60°, achieves the desired value of 10° within 80 ms after issuing the command input. The direction of rotation of the rotary sleeve nozzle and the direction of fluid pressure form a straight angle. Because the fluid pressure does not have a significant impact on the nozzle rotation, it is possible to achieve a quick conversion within 80 ms. Finally, the damping torque of the rotary hydraulic cylinder must be precisely adjusted according to the flexion/extension nozzle angle. The damping torque of the rotary hydraulic nozzle was measured using the Instron and digital torque wrench described in Section 2.7. Figure 10B shows a variation in the cylinder damping torque ranging from 0 to 55.1 Nm at maximum angular velocity (15.7 rad/s) depending on the flexion and extension nozzle angles. Every cylinder has different damping characteristics due to mechanical tolerances and rubber packing performance. As a result, Figure 10A,B indicate that sufficient dampening torque for the amputee's gait cycle may be produced within 80 ms. This suggests that the passive mode of the rotary hybrid prosthesis may adequately assist level ground walking for amputees. The damping coefficient, according to angular velocity, tends to increase at the nozzle angle of 20 degrees but decreases at the nozzle angle of 50 degrees, as shown in Figure 10C,D and Table 3.

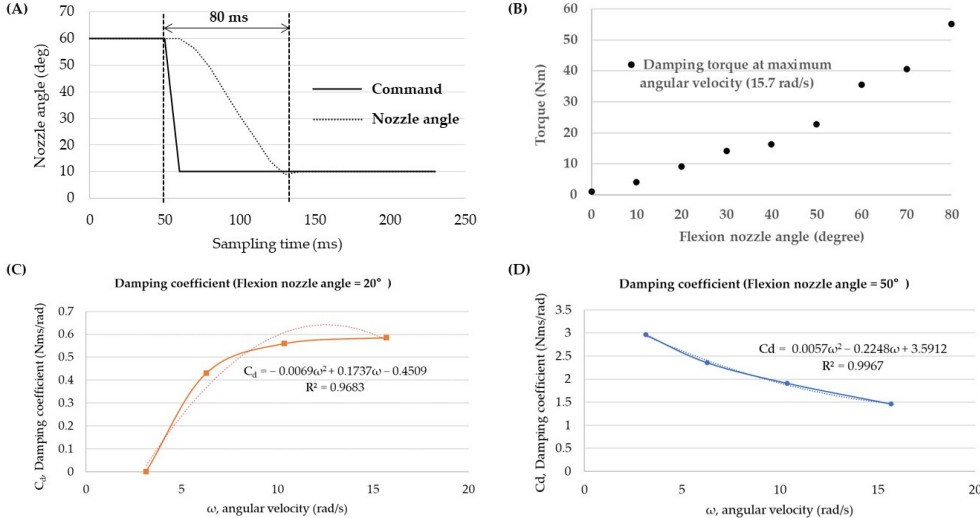

**Figure 10.** Nozzle characteristics test: (**A**) nozzle response test according to shifting nozzle angle; (**B**) cylinder damping torque at maximum angular velocity (15.7 rad/s) according to nozzle angle using the Instron test machine and a digital torque wrench; the damping coefficient according to angular velocity at the nozzle angle of (**C**) 20 degrees and (**D**) 50 degrees.

**Table 3.** The calculation of damping coefficient.

| ω, Angular Velocity (rad/s) | 3.14 | 6.28 | 10.36 | 15.7 |
|---|---|---|---|---|
| $T_d$, Damping torque at flexion nozzle angle of 20 degrees (Nm) | 0 | 2.7 | 5.8 | 9.2 |
| $C_d$, Damping coefficient at flexion nozzle angle of 20 degrees (Nms/rad) | 0 | 0.43 | 0.56 | 0.59 |
| $T_d$, Damping torque at flexion nozzle angle of 50 degrees (Nm) | 9.3 | 14.8 | 19.8 | 22.9 |
| $C_d$, Damping coefficient at flexion nozzle angle of 50 degrees (Nms/rad) | 2.96 | 2.36 | 1.91 | 1.46 |

*3.2. Verification of Active Mode Operation (Feasibility Test)*

The following requirements must be met to achieve the available performance of the rotary hybrid prosthesis in active mode. First, the flow path in the rotary hydraulic cylinder must be entirely locked, such that the rotary cylinder moves as if it were a rigid body. The maximum braking torque of 55.1 Nm is attained when the nozzle is completely 'closed' at 80 degrees, as shown in Figure 10B. This maximum braking torque measurement exceeds 40 Nm, which is a peak torque in the active drive module, indicating that the force required to fix the rotary blade in active mode is adequate. Second, sufficient speed must be achieved to produce a swing phase gait pattern. The amputees that walk fast have a swing phase of fewer than 0.5 s. When the maximum swing flexion angle is set to 60 degrees, an angular velocity greater than or equal to 240 deg/s (40 RPM) is required to provide smooth walking.

In this study, the swing phase walking environment was simulated by setting up an experimental setup, as illustrated in Figure 11A. Figure 11B shows the movement of the prosthetic limb from 0 to 60 degrees in the swing phase with the aid of a digital goniometer. Figure 11C demonstrates how the actual knee angle closely followed the command input with a 30 ms lag during the swing phase in automatic mode. During the test, the prosthesis consumed 4A and 3A of current in the early and terminal swing phase, respectively, which generated the swing motion and deceleration. An increase in current supply allowed the angular velocity to reach 300 deg/s (50 RPM). This prosthesis was designed as an early prototype with a focus on developing a swing phase gait pattern and was optimized for a target speed of 50 RPM and an active torque of up to 40 Nm. The results showed that these goals were successfully achieved. Further details on the maximum active torque test methods and results can be found in the supplementary materials in Figure S1 and Table S1. At the pre-clinical test, the knee angular velocity is set according to the patient's chosen gait speed. By altering the gear ratio of the upper and lower pulleys or the harmonic gear with a high gear ratio, it is possible to construct a prosthesis most appropriate for the active walking of amputees in the future.

*3.3. Clinical Test for a Transfemoral Amputee*

A clinical trial was conducted on rotary hybrid prostheses and involved a transfemoral amputee who wore a prosthesis wearable orthosis during the study. The trial was approved by the IRB (IRB number: KCIRB-2022-0012) and conducted in accordance with standard safety protocols to guarantee the participant's safety. The test subject was a 57-year-old male with a right transfemoral amputation, weighing 67 kg and measuring 172 cm in height. The participant had previously participated in several clinical trials involving various prosthetic limbs, thus possessing a thorough understanding of amputee gait. The hip and knee kinematic data were evaluated using a three-dimensional motion analysis system (Motion Analysis Corp., USA). The 3D gait analysis data acquisition was performed in accordance with the authors' previous study method [23].

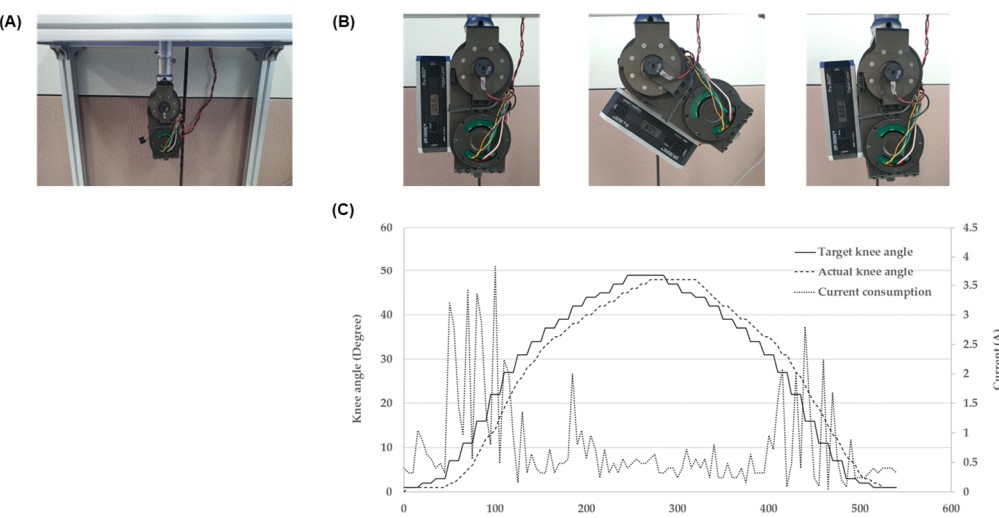

**Figure 11.** (**A**) Measurement test environment for gait speed during swing phase; (**B**) the movement of the prosthesis during swing phase (0° -> 60° -> 0°); (**C**) an actual knee angle and current consumption according to the command input during the swing phase in the automatic mode.

Figure 12A showcases snapshots of the participant walking smoothly while using our prosthesis in passive mode. The knee and hip joint kinematics of the prosthesis during level ground walking are depicted in Figure 12B, alongside those of the healthy knee. During the trial, the stance knee flexion of the prosthetic side was less pronounced compared to the healthy side. Meanwhile, in the swing phase, the knee joint trajectory of the prosthetic side closely mimicked that of the healthy side. In addition, Figure 12C displays quantitative plots of the hydraulic valve angle and ankle moment. The ankle moment sensor, mounted on the lower proximal part of the prosthesis, is utilized to identify the gait phase, and the hydraulic valve angle is adjusted to provide suitable joint resistance for the respective gait phase. The initial flexion damping torque was set to 35.5 Nm in the stance phase and 10 Nm when transitioning to the swing phase. Concurrently, the flexion/extension nozzle angle was set to 60/20° in the stance phase and 20/70° in the swing phase. After training, the flexion/extension nozzle angle in the stance phase was adjusted to 58/20° and 20/65° in the swing phase according to the subject's preference. Our findings indicate that as the subject became more accustomed to the prosthetic limb, the flexion damping torque decreased, and the extension damping torque increased.

Figure 13A illustrates an amputee using our prosthesis while walking on level ground in an active mode, as viewed from the front. The participant's gait was also evaluated in this mode. The knee and hip joint kinematics of the prosthesis during level ground walking are depicted in Figure 13B, along with those of the intact knee. To ensure the safety of the participant, we set an initial low angular velocity (200 deg/s) and a maximum swing flexion angle of 50°. As the subject's gait speed increased throughout the clinical trial, we adjusted the maximum knee angle, reducing it to 46°. The subject initiated swing flexion at 50% of the gait cycle, which was approximately 0.1 seconds faster than their intact side. Our findings indicate that active mode walking is possible when the hydraulic cylinder's flexion/extension nozzle is kept 'closed', preventing any relative movement of the blade. Figure 13C displays quantitive plots of current and knee torque. The highest consumption of current occurred at the start of swing flexion, which generates the swing motion, and at the end of swing extension, which necessitates deceleration. Additionally, a torque sensor mounted on the upper proximal part of the prosthesis is utilized to detect the heel-off event.

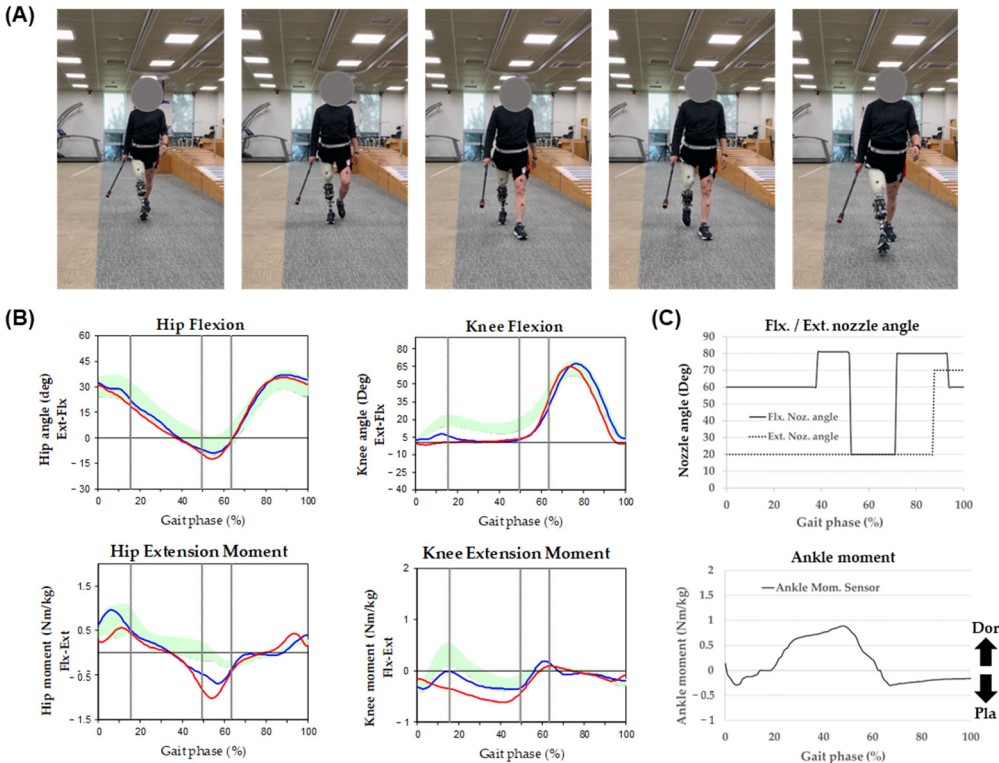

**Figure 12.** Gait analysis in passive mode, (**A**) Snapshots of the amputee's frontal view during walking, (**B**) The knee and hip joint kinematics of our prosthesis during level ground walking using a three-dimensional motion analysis system, (**C**) Quantitative plots of hydraulic valve angle and ankle moment mounted on the prosthesis during level ground walking.

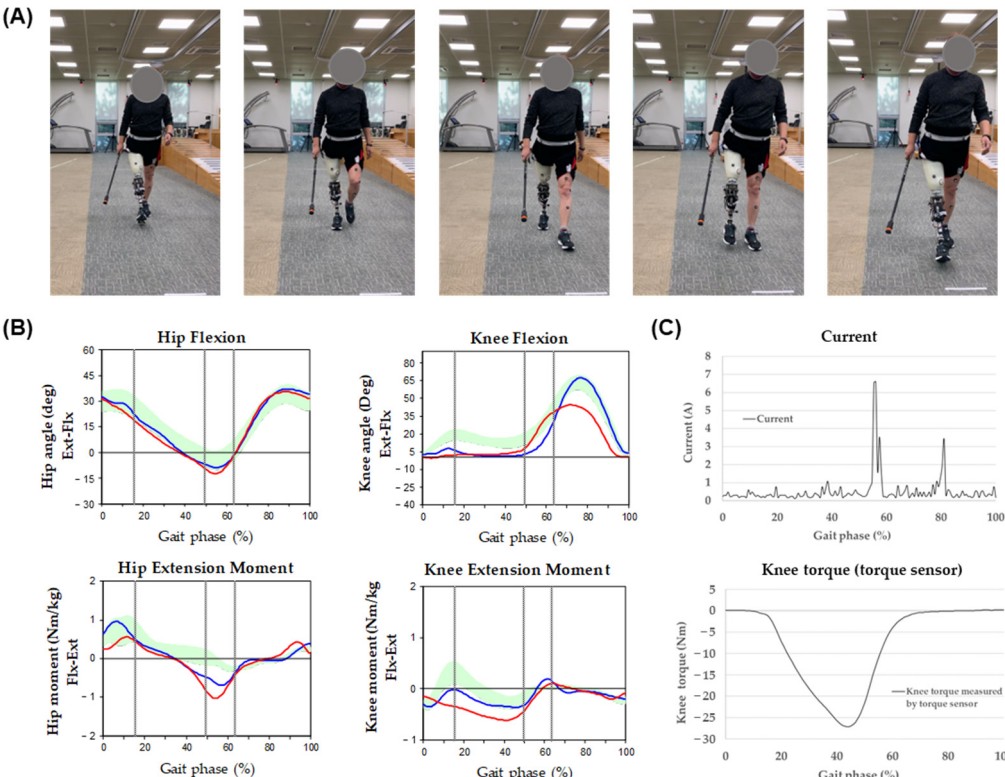

**Figure 13.** Analysis of gait in active mode, (**A**) Snapshots of amputee's frontal view during walking, (**B**) The knee and hip joint kinematics of our prosthesis during level ground walking using a three-

dimensional motion analysis system, (**C**) Quantitative plots of current and knee torque mounted on the prosthesis during level ground walking.

## 4. Discussion

Except during training, most transfemoral amputees walk on level ground or low slopes, and deliberate walking on stairs or steep hills is exceedingly unusual. In the event of level ground walking, normal passive prostheses are sufficient [24–27]. Walking with a high-powered motor that uses much energy in all walking scenarios is, therefore, energy-inefficient. A limited use time of 2–4 h is the main reason why the prosthesis of the purely active method developed in previous research has not been fully commercialized [28]. In this study, a hybrid-type prosthesis that can selectively drive passive and active amputees depending on their gait state or walking terrain by merging a multifunctional rotary hydraulic cylinder with a motor-based active drive module was developed. Table 4 shows the main characteristics of the prosthetic leg developed in this study.

**Table 4.** Mechanical and electrical characteristics of powered prosthesis.

| Item | Value |
| --- | --- |
| Weight (kg) (without battery) | 2.59 |
| Height (mm) | 227 |
| ROM (degree) | 122 |
| Cylinder locking torque (Nm) | 62.1 |
| Active locking torque (Nm) | 65 |
| Active torque (Nm) | 39.4 |
| Swing angular speed (deg/s) | 247.9 |
| Operating mode | 2 (Passive/Active) |
| Operating time (Battery: 7-cell condition) | 120 h over (Passive mode), 6 h (Active mode) |

The most significant feature of this prosthesis is the ability to operate its passive and active modules independently. This decoupling between the modules, results in energy savings as neither mode functions as a load when the other mode is activated. The use of a 24 V, 24,500 mAh battery may allow driving in passive mode for 120 h or longer without needing to recharge. Another significant advantage is that the prosthesis is capable of switching between passive and automated modes, or between stance and swing phase in less than 80 ms. If the transition from the stance phase to the swing phase takes a long time, it might induce anxiety in the patient and may also result in safety issues. This prosthesis may switch modes fast without consuming significant power through control of the rotary sleeve nozzle in the hydraulic cylinder. In passive walking, the control method of commercial MPK, which adjust joint resistance with hydraulic damping force like this study, is more advantageous in terms of energy consumption than the gait control of an typical active prosthesis. However, because the rotary hydraulic cylinder developed in this study does not possess a suitable mechanism to return the blade to its original position, it can only return to its original position by the femoral movement of the amputees. In our previous study, a rotational spring was used to address this issue, but it was not incorporated into the model owing to its difficulty in handling heavy loads during flexion and long-term durability issues. In future studies, we will aim to enhance the return mechanism with a dual rotational spring. Please note that the development of rotary actuator is still in the verification stage, further clinical trials will be required to determine its applicability for amputees in the future.

## 5. Patents

As a result of this study, a Korean patent (KR1024673780000) was registered.

**Supplementary Materials:** The following supporting information can be downloaded at: https://www.mdpi.com/article/10.3390/act12030118/s1, Figure S1: (A) Experimental setup to measure the maximum active torque on the test bench. (B) Free body diagram based on the center of the knee joint in test bench, (C) The babel lift test: the ability of the active drive to provide active torque to the joint; Table S1: The parameters of the active torque measurement.

**Author Contributions:** Conceptualization, H.S., M.J. and J.P.; methodology, H.S. and S.J.; software, H.L.; validation, H.S., J.P. and H.L.; formal analysis, H.S. and S.P.; investigation, H.S.; resources, S.P.; data curation, H.S. and H.L.; writing—original draft preparation, H.S.; writing—review and editing, H.S.; visualization, H.S. and H.L.; supervision, S.P.; project administration, S.P.; funding acquisition, S.P. All authors have read and agreed to the published version of the manuscript.

**Funding:** This work was supported by the Korea Medical Device Development Fund grant funded by the Korean government (the Ministry of Science and ICT, the Ministry of Trade, Industry and Energy, the Ministry of Health & Welfare, the Ministry of Food and Drug Safety) (Project Number: 1711139032, KMDF_PR_20200901_0206). This research was supported by a grant from the Korea Health Technology R&D Project through the Korea Health Industry Development Institute (KHIDI), funded by the Ministry of Health & Welfare, Republic of Korea (grant number: HJ22C0003).

**Institutional Review Board Statement:** KCIRB-2022-0012.

**Informed Consent Statement:** Informed consent was obtained from all subjects involved in the study.

**Data Availability Statement:** Not applicable.

**Conflicts of Interest:** The authors declare no conflict of interest. The funders had no role in the design of the study; in the collection, analyses, or interpretation of data; in the writing of the manuscript, or in the decision to publish the results.

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
