# Peer review of "Selective Passive/Active Switchable Knee Prosthesis Based on Multifunctional Rotary Hydraulic Cylinder for Transfemoral Amputees"

_actuators, doi:10.3390/act12030118_

Round 1

Reviewer 1 Report

Please find remarks in an attached file.

Reviewer 2 Report

While the authors’ introduction of a hybrid hydraulic knee contains merits, the authors should be clearer in differentiating their contributions compared to the state-of-the-art in the design of both semi-active/hybrid and powered prostheses. Additionally, more details on the experimental results are needed. I have included my comments below:

Major comment 1: Please highlight your scientific contributions related to prosthesis design

1)     The authors should acknowledge and cite the more recent state-of-the-art in robotic knee prostheses across different architectures (powered/hybrid/semi-active). I suggest at least the four below:

 [1*] Bartlett, Harrison Logan, et al. "Design and Assist-As-Needed Control of a Lightly Powered Prosthetic Knee." IEEE Transactions on Medical Robotics and Bionics 4.2 (2022): 490-501.

 [2*] Culver, Steve C., Leo G. Vailati, and Michael Goldfarb. "A Power-Capable Knee Prosthesis With Ballistic Swing-Phase." IEEE Transactions on Medical Robotics and Bionics 4.4 (2022): 1034-1045.

 [3*] Tran, Minh, et al. "A lightweight robotic leg prosthesis replicating the biomechanics of the knee, ankle, and toe joint." Science Robotics 7.72 (2022): eabo3996.

 [4*] Guercini, Lorenzo, et al. "An Over-Actuated Bionic Knee Prosthesis: Modeling, Design and Preliminary Experimental Characterization." 2022 International Conference on Robotics and Automation (ICRA). IEEE, 2022.

2)     The authors should make clear the scientific contributions of their particular design compared to the state-of-the-art. Does their prosthesis offer any functional/clinical advantages compared to existing designs, such as reduced weight/size or improved performance?

Major comment 2: Please provide more quantitative data for experimental results:

3)     Please report the achievable range of damping coefficient at the knee joint due to the controlled opening/closing of the hydraulic valves

4)     To complement figure 11(b), please show the desired and actual knee position trajectories that correspond to the commanded current, and comment on the tracking performance.

5)     Instead of video snapshots of the walking bypass experiments as in Figures 12 and 13, please include quantitative plots of those experiments. The plots should contain joint position, joint torque, and other useful information (hydraulic valve position, active motor power, etc.)

6)     Additional experiment requested: Please provide some test that shows the ability of the active drive to provide active torque to the joint, which is closer to its full expected capacity (40 Nm) than simply swinging the knee. In my opinion, either an experiment on the bench or with the bypass is ok.

Reviewer 3 Report

The article presents a Passive/Active Switchable Knee Prosthesis based on a novel mechanism.

The formal analysis and rationale are presented with great clarity.

The research lacks a comprehensive dynamic analysis demonstrating the prothesis's possible trajectories and movements.

In addition, a walking analysis is performed to evaluate the performance of the suggested prosthesis. However, the gait analysis should be conducted with greater attention to detail with an in deptht discussion.

Please use these papers as a guide to add the missing section that will increase the quality of your paper:

https://doi.org/10.3390/app10186168

https://doi.org/10.1007/978-3-319-09858-6 47

After these improvements the paper in my opinion can be published as it is interesting an in line with the journal topics.

Round 2

Reviewer 2 Report

The authors have sufficiently addressed all of my comments. I greatly appreciate the addition of amputee and high-torque experiments.